# Determining the Efficiency of Small-Scale Propellers via Slipstream Monitoring

**Jaan Susi \*, Karl-Eerik Unt and Siim Heering**

Department of Aeronautical Engineering, Estonian Aviation Academy, Lennu 40, Reola, 61707 Kambja Parish, Tartu County, Estonia; karl-eerik.unt@eava.ee (K.-E.U.); siim.heering@eava.ee (S.H.)
\* Correspondence: jaan.susi@eava.ee

**Abstract:** A large part of small-sized UAVs that are used for surface scanning, video- and photography, or other similar applications are of the multirotor type. These small aircraft perform mainly in hovering or nearly hovering flight mode, and the endurance of these vehicles depends greatly on the efficiency of their motors and the aerodynamic efficiency of their thrust-generating systems, including propellers, ducted fans, etc. Propellers may therefore work in different regimes: in a regime where the propeller performs work to move the vehicle through the air, and the static or hovering regime, in which standing air is accelerated. In both cases, the concept of efficiency can be used to describe the propeller's performance. There have been several previous studies on static and advancing propellers' performances. In these studies, when determining the efficiency of a static propeller, the thrust and power coefficients are most commonly compared to evaluate the propeller's performance. Sometimes, the inducted velocities are calculated via the momentum theory. As small-scale propellers work on very low Reynolds (Re) numbers below 500,000, the flow type transition and boundary layer separation make it very hard to predict the actual efficiency of the propellers in static mode. Therefore, the aim of this paper is to introduce a method to determine the static efficiency of small-scale propellers directly and empirically via a comparison between the output and input power, wherein the output power is determined via the measured thrust and mean induced velocity. The used method combines thrust, torque, and angular velocity measurements with slipstream monitoring. The performed tests showed a decrease in efficiency, with the Re number rising in spite of the rising values of the thrust coefficient. This study led to two main conclusions: thrust and power coefficients are not always the key parameters to determine the efficiency of a propeller; the role of the Re number in the propeller's efficiency is not yet clear and requires further investigation. The presence of Re number effects has been proven in numerous works, but the impact of those effects seems to not be as trivial as the claim that the lower the Re number, the weaker the propeller's performance.

**Keywords:** propeller efficiency; applied aerodynamics; small-scale propellers; multirotors

## 1. Introduction

The number of unmanned aerial vehicles (UAVs) used worldwide continues to increase, and this leads to problems in the efficiency and environmental friendliness of these devices. More and more UAVs are used not only as remote control devices for entertainment but also for professional use for civil and military purposes, such as surveillance, photography, and transporting small cargo loads. A large part of UAVs are of the multirotor type and perform mainly in hovering or nearly hovering flight mode and are powered by electric motors and onboard batteries. The endurance of any aerial vehicle depends greatly on the efficiency of the aerodynamic layout of the vehicle and on the aerodynamic efficiency of its thrust-generating systems (propellers, ducted fans, etc.). The efficiency of a UAV is also relevant from the point of view of environmental protection, as more efficient devices use less fuel or prolong the recycling period of onboard batteries. In multirotor-type

small UAVs, thrust is generated with small-scale propellers, often working at Reynolds numbers in the region well below 500,000. The Reynolds number is considered critical for aerodynamic wings and propeller blades. The definition of the term "Small Scale Propeller" is determined by the fact of whether the main parts of the propeller (as a rule, with the blade sections occupying nearly 75% of the propeller radius) are working either at lower or higher Reynolds numbers than the critical Reynolds number. Propellers working at Reynolds numbers lower than the critical Reynolds number are often called small-scale propellers and are the propellers investigated in the present paper (with Reynolds numbers between 70,000 and 200,000). The problem is that this region lies between the two modes of air flow: streamline flow and turbulent flow. Investigations have shown that with the Reynolds number decreasing from a value of 500,000, the efficiency of a propeller also decreases due to the transition between flow types and the earlier boundary layer separation from the propeller blades in the streamline (laminar) flow mode [1–3]. It is often difficult to predict the performance of small-scale propellers [4–6]. This is understandable due to the uncertainty in determining the flow type in this region of Reynolds numbers. Propellers may work in different regimes. The two basic regimes are the propelling regime, wherein the propeller performs work to move the vehicle through the air, and the static or hovering regime, wherein the propeller performs work to induce a slipstream, i.e., to accelerate standing air. In both cases, one can use the concept of efficiency to describe the propeller's performance. Several studies have investigated both the static and advancing performance of small-scale propellers [7–12]. In studies on static performance, the concept of propeller efficiency has not been defined in the usual sense of the word. The problem is that useful power or output power has not been measured empirically, and only the thrust and power coefficients have been compared to evaluate a propeller's performance. Sometimes the ratio of those coefficients is used to evaluate the performance of a propeller. Thrust is the only output characteristic in these cases. The induced velocities for a propeller may be calculated via the momentum theory [13], but for the reasons mentioned above, these calculations may not have enough precision at low Reynolds numbers. This leads to a need for the direct measuring of the induced velocity to define the static efficiency of a propeller via that parameter. The aim of the present paper is to introduce a method to determine the static efficiency of small-scale propellers directly and empirically via a comparison between the output and input powers, whereby the output power is determined via the measured thrust and mean induced velocity. This enables us, with no reservations, to use the term "efficiency" for static regimes of propellers. This approach is quite different from previous similar works wherein only the coefficients of thrust and power were compared to characterize the performance of propellers working in the static regime. The proposed method enables us to quickly and effectively compare the efficiencies of different propellers at different Reynolds numbers. In addition, slipstream monitoring provides the possibility to determine many aerodynamic parameters of propeller blades and acquire information about the behavior of the slipstreams.

## 2. Materials and Methods

When speaking about the efficiency of any kind of power-transforming system, it is usual to compare the output power and input power. This is carried out using the ratio of the output power to the input power, which is often expressed as a percentage. For propellers, the input power is defined as

$$P = M \cdot \omega. \tag{1}$$

The definition of output or useful power depends on the working mode of the propeller. In the case of an advancing propeller, the propulsive efficiency is defined via the airspeed of the vehicle or the velocity of the freestream flow:

$$\eta = \frac{T \cdot v_0}{P} . \tag{2}$$

The thrust and input power of a propeller can be presented as

$$T = C_T \cdot \rho \cdot \omega^2 \cdot D^4, \ P = C_p \cdot \rho \cdot \omega^3 \cdot D^5. \tag{3}$$

The propulsive efficiency is often defined via the advance ratio:

$$\eta = J\frac{C_T}{C_p} \ , \ J = \frac{v_0}{\omega R}. \tag{4}$$

In both cases, it is easy to measure the parameters necessary to determine the efficiency. In the static case, i.e., when the freestream velocity equals zero, the output (induced) power of the propeller and the efficiency are defined as

$$P_i = T, \ \eta = \frac{P_i}{P}. \tag{5}$$

The mean induced velocity $\overline{v_i}$ cannot be correctly measured directly but can be derived from the results of slipstream monitoring which, in our case, means measuring all three components of the induced velocity. In theory, the induced velocity is the velocity of air in the plane of rotation. In empirical investigations, the induced velocity should be measured as closely as possible to the trailing edge of the propeller blade. Herein, when carrying out the experiment, one must consider the possibility of propeller deformation under the resulting forces of inertia and thrust. This may lead to the need to change the axial position of the measuring probe depending on the propeller's rotating speed. The axial distance of the airspeed measurement must not be far from the trailing edge of the propeller blade to minimize mass (or volumetric) rate losses in the slipstream that will grow with distance. The mean induced velocity is calculated using the measured volumetric rate and full propeller diameter. The simplest model of a slipstream (in the case of an ideal propeller) predicts the contraction of the slipstream according to the momentum theory [13]. The contraction takes place due to static pressure differences inside and outside of the slipstream. The simplest approach in slipstream theory says that the contraction of a slipstream ends and the full pressure balance is achieved when the slipstream velocity is doubled in comparison with the induced velocity. Herein, we see that the simplest model of a slipstream predicts the presence of the radial flow component of the slipstream toward the axis due to contraction. In practice, the situation is more complicated as the slipstream is rotating. This means that in any location inside a slipstream, we have three components of airflow velocities (Figure 1): the axial, the radial, and the tangential components.

In propeller efficiency, only the axial component plays a positive role in generating useful power as it is directed in the thrust line. The tangential component is caused by the rotation of the slipstream. The latter is caused by the torque of the propeller and, furthermore, by the drag of the blade sections. The drag of the blade sections depends on their airfoil parameters, angle of attack, flow separation from the blade back, etc. It also depends not only on the angle of attack but also on the blade angle itself (see Figure 2). The higher the blade angle, the less it contributes to the lift force of a blade section into the thrust and the more it contributes to the drag moment (torque). This effect is present even when the blade section has the most effective angle of attack.

The rotation of the slipstream that creates the velocity component $v_t$ has a direct impact on the remaining two velocity components of the slipstream and on the propeller's performance in general. First, let us suppose that the whole slipstream rotates with an angular velocity $\frac{v_t}{r}$. The centrifugal acceleration on the slipstream surface is then $\frac{v_t^2}{r}$ and zero on the axis. In an air column reaching from the slipstream surface to the axis (Figure 3), the radial centrifugal pressure $\frac{\rho v_t^2}{2}$ built by the centrifugal force is determined by one half of the centrifugal acceleration on the slipstream surface.

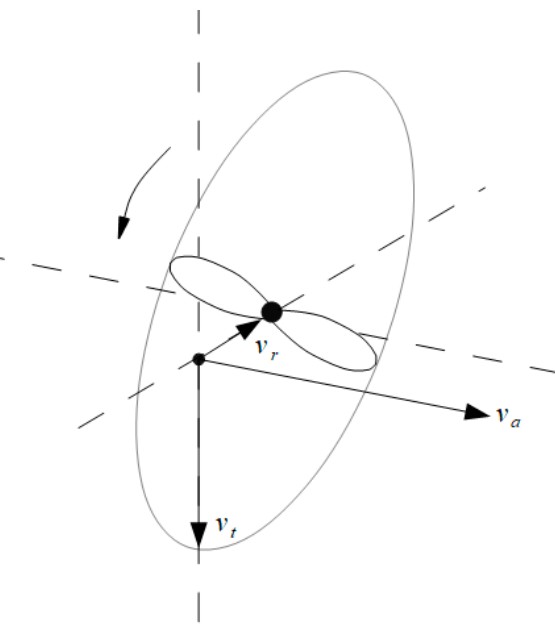

**Figure 1.** The three components of air velocities in a slipstream: $v_a$—axial component, $v_t$—tangential component, and $v_r$—radial component.

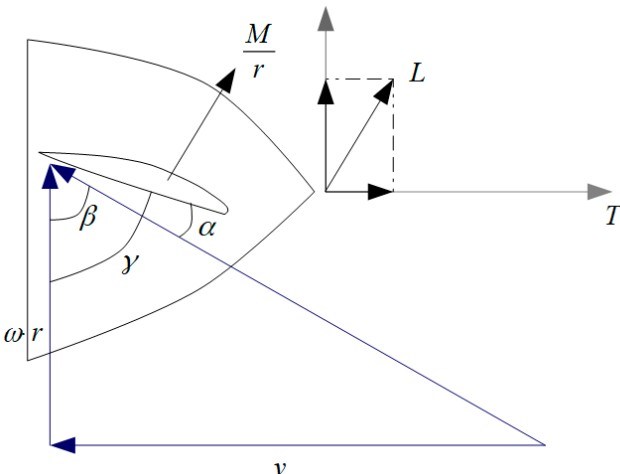

**Figure 2.** The role of lift force of a blade section in creating thrust and torque: *L*—lift force of a blade section, *M*—drag moment of blade section, *r*—distance of blade section from the propeller axis, *T*—thrust generated by blade section, $\omega$—angular velocity, $v_a$—axial component of the airflow, $\gamma$—blade angle, $\beta$—helix angle, and $\alpha$—angle of attack.

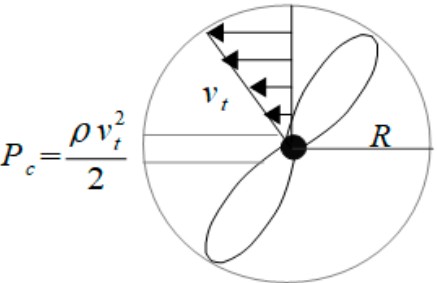

**Figure 3.** The buildup of the centrifugal pressure in a slipstream: $P_c$—centrifugal pressure, $\rho$—air density, $v_t$—tangential component of air velocity and *R*—propeller radius.

This centrifugal pressure counteracts the static pressure difference in and outside the slipstream and consequently counteracts the slipstream contraction. In the case of high torque (due to too high blade angles and/or low Reynolds number effects), the slipstream may even be expanding. Indeed, by increasing the blade angle to high values, the propeller begins to work more and more as a rotor of a centrifugal compressor, forcing the slipstream to expand and decreasing the role of the axial flow component. In the case of a contracting slipstream, the dynamic pressure (that is, the density of the kinetic energy in the flow) rises and provides feedback to the propeller blade faces in the form of additional pressure. That additional pressure on the blades has a direct impact on the thrust and thus on the efficiency of the propeller. The smaller the effect of the contraction is, the less thrust there is, and thus, the efficiency will decrease. However, the assumption of the angular velocity being constant for the whole slipstream results in a simplified or first-step model. In a more precise model, we must assume counteraction between the velocity components $v_r$ and $v_t$. This means that if the slipstream is contracting, there will also be a direct influence of $v_r$ on $v_t$ along with the inverse effect caused by the centrifugal force. This influence is often called the Coriolis effect. The conservation of angular momentum in a slipstream entails that $v_t \cdot r = const$ for any radially moving air parcel. From this, the Coriolis acceleration is

$$\frac{dv_t}{dt} = \frac{const}{r^2} \cdot \frac{dr}{dt} = -\frac{v_t}{r} \cdot v_r \tag{6}$$

The Coriolis effect counteracts the slipstream contraction because it raises $v_t$, and thus, the centrifugal pressure increases in the central parts of the contracting slipstream in comparison with the previous model presented in Figure 3. In the case of an expanding slipstream, the Coriolis effect counteracts the expansion as well, because then $v_t$ slows down and decreases the centrifugal pressure. Neither of the models described above considers the energy dissipation in a slipstream due to viscosity and turbulence. From this comes the reason slipstream measurements, if carried out, must be performed as closely as possible to the trailing edge of the propeller blades to minimize the effect of slipstream energy losses.

Thus, one can conclude that the behavior of a slipstream plays an important role in the efficiency of a propeller and offers several parameters to be investigated to improve a propeller's performance. Herein, the most important aspect is to gather calibrated measurements of all three velocity components of a slipstream flow. This can be performed with the help of special equipment containing a thin complex multi-hole tube as a sensor to measure the combinations of total and static Bernoulli pressures in the slipstream. All three components are then distinguished, and the radial distributions of all components can also be determined. To determine the propeller efficiency, the value of the mean induced velocity must be used. The latter is determined using the radial distribution of the measured axial velocity component (Figure 4).

First, the volumetric rate of the slipstream is determined. If the radial distribution of the axial component is presented as $v_a(r)$, the volumetric rate is calculated as

$$v_v = 2\pi \int_0^R v_a(r) \cdot r \cdot dr \,. \tag{7}$$

The mean induced velocity and the mass rate are then determined as

$$\overline{v_i} = \frac{v_v}{\pi R^2} \,, \, v_m = \rho \cdot v_v \,. \tag{8}$$

The propeller efficiency is now calculated using the measured values of thrust, torque, and angular velocity via Formula (5). In the case of advancing movement wherein the propeller efficiency is defined using the freestream air velocity or advance ratio, efficiency strongly depends on the blade angles (geometric pitch) and freestream velocity. For propellers working in the static regime, we should assume the existence of an optimal blade angle radial distribution to ensure the most effective angle of attack for the blade sections.

These blade angles, however, may depend on the propeller planform, airfoil, Reynolds number, etc., and can also be hardly predictable using simulations. In the present work, the radial distribution of the angle of attack can be determined from the experiment in which the airflow velocity components $v_a$ and $v_t$ are measured. For a blade section, according to Figure 2, the angle of attack is determined as the difference between the blade angle $\gamma$ and the helix angle $\beta$. In the three-dimensional case, one must also consider the radial velocity component $v_r$. Firstly, the effective helix angle and effective chord length must be defined:

$$\beta_{eff} = arcsin \frac{v_a}{v_T} \, , \, d_{eff} = \mathrm{d} cos \, \varphi. \tag{9}$$

were $\varphi$ is the measured azimuthal angle of the airflow, $v_T$ is the total airspeed over a blade section, and d is the measured chord length.

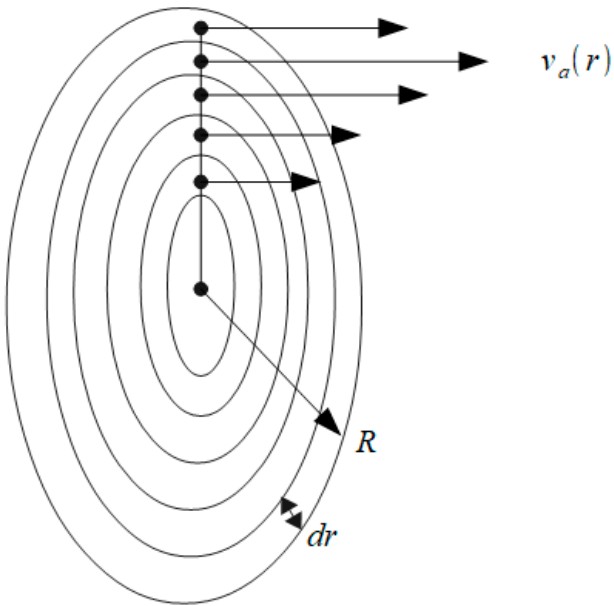

**Figure 4.** Determining the volumetric rate and mean induced velocity: $v_a(r)$—radial distribution of axial component, $R$—propeller radius, and $dr$—distance between measuring points.

For the effective blade angle and the angle of attack, one must consider that these parameters depend on the blade planform and twist geometry. Herein, we can use the simplified approach based on the assumption that the planform and blade angle do not change rapidly along the propeller radius. Under these circumstances, we derive formulae for those parameters as

$$\gamma_{eff} = arcsin \, (sin \, \gamma \cdot cos \, \varphi \,) \, , \, \alpha = \gamma_{eff} - \beta_{eff}. \tag{10}$$

Slipstream monitoring also provides an empirical way to determine the lift and drag coefficients for different propeller blade sections at particular Reynolds numbers and angles of attack. Hence, if a parcel of air interacts with a blade section, the air parcel will then obtain two additional velocity components after that interaction: $v_L$ as the downwash velocity and $v_D$ as the drag velocity (Figure 5).

Supposing the mass rate of the air interacting with the airfoil section is $\frac{dm}{dt}$, the lift and drag forces for that particular airfoil section will then be expressed as

$$L = \frac{dm}{dt} v_L, \, D = \frac{dm}{dt} v_D. \tag{11}$$

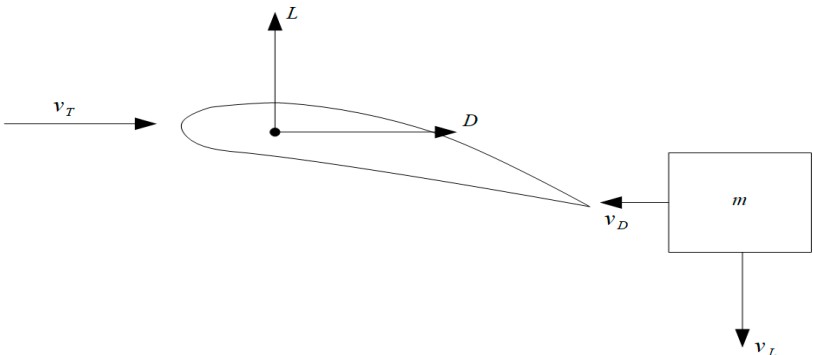

**Figure 5.** Downwash and drag velocities of a parcel of air with mass $m$ after interaction with an airfoil section: $v_T$—total airspeed, $L$—lift force, $D$—drag force, $v_L$—downwash velocity, and $v_D$—drag velocity.

On the other hand, according to the conventional approach,

$$L = C_L \frac{\rho v_T^2}{2} S, \ D = C_D \frac{\rho v_T^2}{2} S, \tag{12}$$

where $v_T$ is the total airspeed over the blade section, and $S$ is the reference area. In the present context, we can express that

$$\frac{dm}{dt} = \rho v_T S \tag{13}$$

and from this,

$$C_L = \frac{2v_L}{v_T}, \ C_D = \frac{2v_D}{v_T} \tag{14}$$

The total airspeed over a blade section is calculated considering all the slipstream flow components:

$$v_T = \sqrt{(\omega r - v_t)^2 + v_a^2 + v_r^2} \tag{15}$$

The velocities induced by the blade section, $v_L$ and $v_D$, are calculated from the measured values of $v_a$, $v_t$, and $v_r$:

$$v_L = v_a \cos \beta_{eff} + v_t \sin \beta_{eff} \ , \ v_D = v_t \cos \beta_{eff} - v_a \sin \beta_{eff}. \tag{16}$$

It must be mentioned that the lift coefficient calculated via Formula (14) is not directly valid for a rotating propeller, because it is derived for an airfoil section free from external influences. In the case of a rotating propeller, one must consider the increased pressure on the blade faces caused by the contracting slipstream. The contracting slipstream achieves some radially averaged maximum velocity value $\overline{v_{max}}$ that is not measured but can be found via measured parameters as

$$\overline{v_{max}} = \frac{T}{v_m} = \frac{T}{\rho \pi R^2 \overline{v_i}} \ . \tag{17}$$

The pressure on the blade faces is directly proportional to $\overline{v_{max}}$ and so is the lift coefficient. As in Formula (14), only the values of axial velocity $v_a$ were used, and the calculated lift coefficient is valid only for a non-contracting slipstream. Otherwise, the effective lift coefficient for a blade section can be found by multiplying the lift coefficient found via (14) by the speed ratio:

$$C_{Leff} = C_L \frac{\overline{v_{max}}}{\overline{v_i}}. \tag{18}$$

This situation has some common features with the ground effect, whereby an airfoil section reaches higher values of lift coefficient at lower values of angle of attack due to pressure changes between the blade face and back caused by external influences on the airflow. In calculating the Re number for a blade section, the total airspeed and effective chord length must be used.

### 3. Test Stand and Measurement Technology

The test stand contains two main elements: the fixed propeller post and the movable airflow sensor post (Figure 6). The air data probe is driven using two stepper motors and jackscrews. The main movement of the air data probe is radial to scan the slipstream from its center up to the propeller radius.

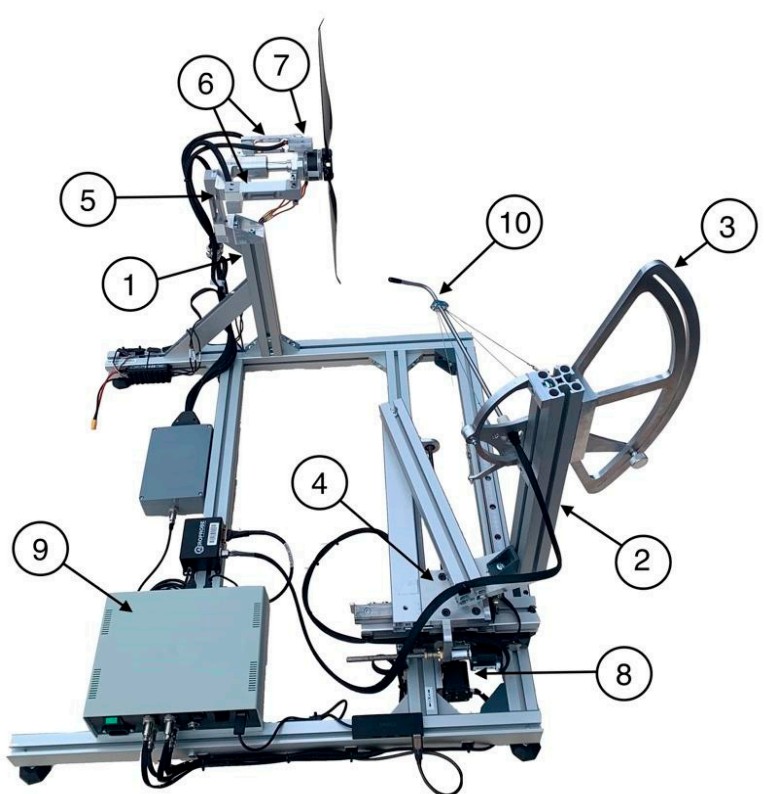

**Figure 6.** The test stand. 1—propeller post, 2—airflow sensor post, 3—initial angle of attack setting system, 4—initial sideslip angle setting system, 5—thrust sensor, 6—torque sensors, 7—RPM sensor, 8—two-coordinate scanning mechanism, 9—control unit, and 10—air data probe with anti-vibration cables.

The axial movement of the airflow sensor indicates the necessary scanning trajectory of the sensor tip, depending on the shape of the propeller blades and the possible flexing of the blades due to propeller thrust. The probe is secured with thin steel cables to avoid possible vibrations of the probe tube in the airflow. The torque and thrust sensors, as well as the optical tachometer, are mounted on the propeller post. As the air data probe can detect the angle of attack and sideslip angle in a region between +21 degrees and −21 degrees, the initial angle setting systems were incorporated into the airflow sensor post to widen the range of measured airflow directions. An external strobe light source can be used to visually monitor the propeller rotation and detect possible blade vibrations and the stability of the angular velocity. A high-speed camera can also be used to photograph the rotating propeller to determine the exact amount of blade bending that is necessary to determine the scanning trajectory of the sensor tip at a constant distance from the trailing edge of the blade (Figure 7).

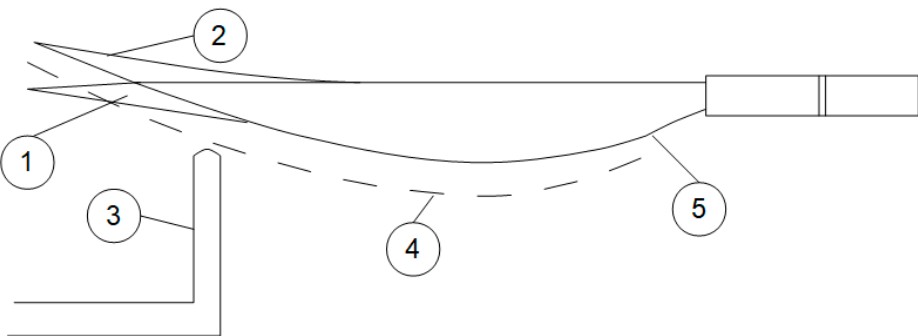

**Figure 7.** Determining the scanning trajectory at a constant distance from the trailing edge of a propeller blade. 1—standing blade (side projection), 2—rotating blade flexing under thrust, 3—airflow sensor tube, 4—scanning trajectory, and 5—trailing edge.

One Zemic L6D single-point load cell is used to measure torque, and another is used to measure thrust. The cells can easily be changed according to the necessary measuring range. The cells have 0.014% precision, and if calibrated according to force, torque, and thrust, these can be measured with an average relative error of 0.1%. With an optical tachometer, it is possible to stabilize the angular velocity with a precision of 0.05 rad/s, which, in the case of a propeller working at 3000 RPM, produces a relative error of 0.02% for angular velocity. This makes it possible to calculate the thrust and power coefficients with relative errors of approximately 0.5%, including an air density error that is estimated to be 0.1%. The slipstream is monitored with the Aeroprobe Micro Air Data System using a calibrated multihole air data probe that detects the full airspeed and two perpendicular angles: the angle of attack and the sideslip angle. The absolute error of the airflow velocity measurement is 1 m/s in terms of the absolute calibration of the sensor. The relative changes in the airflow velocity can be detected with a precision of 0.2 m/s without any difficulties. In the case of a typical radial distribution of the induced velocity with a peak value of 25 m/s, the relative error for the mean induced velocity is estimated to be 5% at a 95% trust level. This means that when calculating the propeller efficiency, the error of that parameter is mainly determined by the error of the induced velocity. Thus, the expected relative error of the efficiency will also be approximately 5% from the calculated efficiency, not from the input power, which is considered to be 100%. For instance, if the propeller efficiency is determined to be 70%, the full-scale relative error will be 3.5%. The changes in the efficiency can be detected with approximately 5 times higher precision, i.e., with an error of approximately 0.7% on the full (100%) scale.

## 4. Results

The computer-controlled measurement process includes the direct measuring of the propeller angular velocity, torque, and thrust at the maximum possible stability of those parameters. The slipstream scanning is performed by moving the five-channel sensor tube radially, starting from the center of the slipstream and following the predicted trajectory to keep the measuring distance from the propeller trailing edge constant. It was found that a distance of approximately 4 to 5 mm from the trailing edge is sufficient from the point of view of both accuracy and safety. The measuring device enables the radial scanning of the slipstream with a step of 1 mm, but taking into account the diameter of the measuring probe (0.25 of an inch), the measuring step was set to be 3 mm, corresponding to one-half of the probe's diameter. In this case, all the measurements within a 3 mm step are averaged in the output data. As the propeller slipstream is turbulent, there may be considerable temporal fluctuations in the parameters measured with the air probe. These fluctuations depend very much on the propeller's configuration and its working regime. To obtain temporally averaged results, the number of radial scanning sweeps was increased depending on the propeller type and speed of rotation. Usually, it was enough to average the results of 10 radial sweeps to obtain acceptably smooth radial distributions of the output parameters.

The dependence of the measured results on the air pressure and temperature was not investigated in this work, but as the measuring procedure is quite quick, there was no problem carrying out all the measurements during a short period of time, which means at a constant air pressure and temperature. The air probe measures the module of the total airspeed vector in the slipstream that we call the total slipstream velocity and two angles, the one that the manufacturer calls the angle of attack and the other being the sideslip angle. In terms of the spherical coordinates, the first of those is called the polar angle and the second is called the azimuthal angle. The directly measured parameters in a test are torque, thrust, angular velocity, total slipstream velocity, polar angle, and azimuthal angle. The technical parameters of the propeller such as diameter, radial distribution of blade angle, and airfoil chord are also measured directly and used as input data in further calculations of different parameters. The calculated parameters may be divided into two groups, the integral and differential ones. The integral parameters characterize the whole propeller and contain the input power $P$, induced power $P_i$, mean induced velocity $\overline{v_i}$, volumetric rate $v_v$, mass rate $v_m$, and propeller efficiency $\eta$, thrust coefficient $C_T$, and power coefficient $C_P$. Various differential parameters include those with a radial distribution and are calculated via the formulae discussed above. They can be presented in the form of table columns and/or graphs as needed and are listed as follows: axial velocity $v_a$, tangential velocity $v_t$, radial velocity $v_r$, effective blade angle $\gamma$, effective helix angle $\beta_{eff}$, angle of attack $\alpha$, total airspeed over a blade section $v_T$, downwash velocity induced by a blade section $v_L$, drag velocity induced by a blade section $v_D$, lift coefficient and effective lift coefficient of a blade section $C_{Leff}$, drag coefficient of a blade section $C_D$, effective aerodynamic chord of a blade section $d_{eff}$, and Reynolds number for a blade section. The three components of the slipstream velocity—axial velocity $v_a$, tangential velocity $v_t$, and radial velocity $v_r$—are calculated from the directly measured air probe data by transferring the measured spherical coordinates into a Cartesian system.

First, a widely used multirotor-oriented Tarot 1755 propeller was tested. It is a carbon fiber propeller that is 17″ in diameter with a 5.5″ geometric pitch. It is quite thin (with a relative thickness of 0.75 R, approximately 4.3%) but has quite a high camber (with a relative camber of 0.75 R, approximately 5.3%) airfoil. It also has blade tip modifications such as propeller tips bent toward the blade back (approximately 15 degrees near the tip). The characteristic views of the propeller are presented in Figure 8.

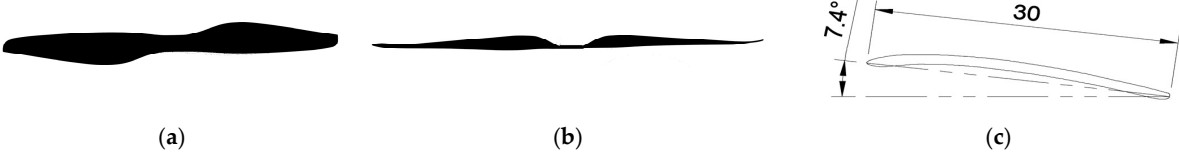

(a)　　　　　　　　　　　　　　　　(b)　　　　　　　　　　　　　　　　(c)

**Figure 8.** Planform (**a**), 83-degree side projection (**b**), and 0.75 R airfoil shape (**c**) of a Tarot 1755 propeller.

The measured radial distribution of the blade angle and airfoil chord length are presented in Figure 9.

The propeller was tested at angular velocities between 220 rad/s and 600 rad/s. It generates a Reynolds number ranging from $7.0 \cdot 10^4$ to $2.0 \cdot 10^5$ for the blade section at 75% of the propeller radius. Monitoring was carried out for five different angular velocities including the minimum and maximum values. The radial distributions of the three slipstream velocity components at the minimum and maximum angular velocities are presented in Figure 10.

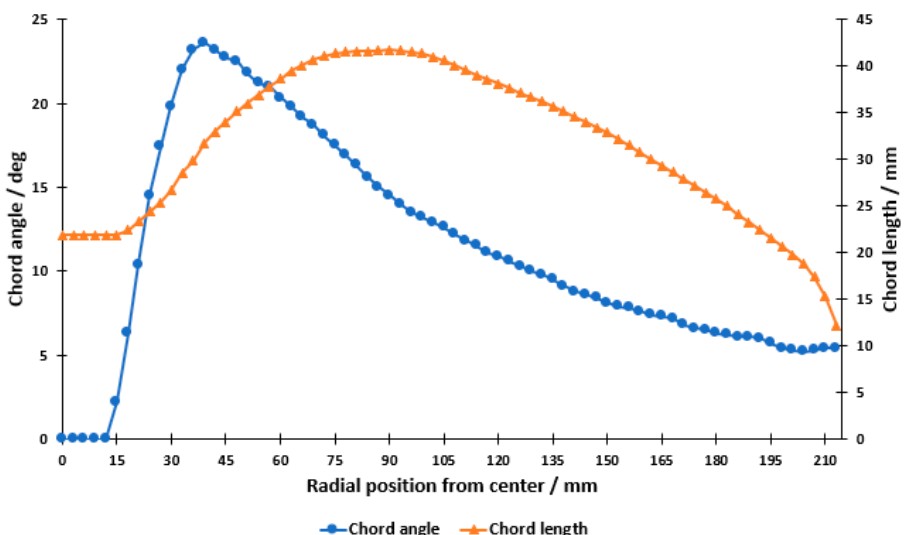

**Figure 9.** Radial distribution of blade angle and chord length of a Tarot 1755 propeller.

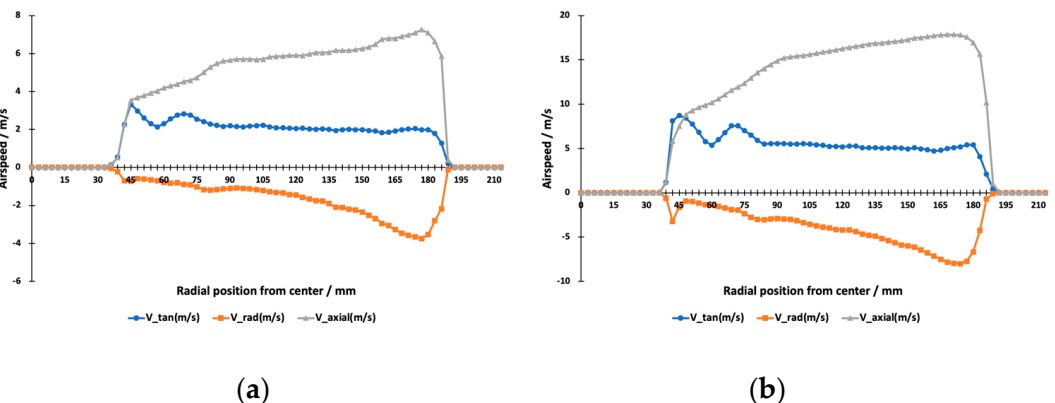

(**a**) (**b**)

**Figure 10.** Radial distributions of $v_a$, $v_t$, and $v_r$ at approximately 2000 RPM (**a**) and approximately 6000 RPM (**b**). The radial velocity toward the axis is considered negative.

In Figure 11, the dependence of the thrust coefficient, power coefficient, and efficiency on the angular velocity is presented.

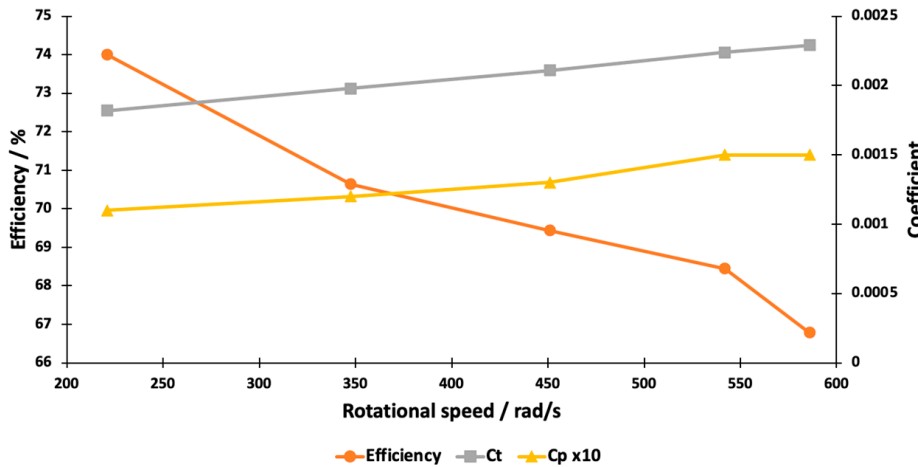

**Figure 11.** The dependence of $C_T$, $C_P$, and $\eta$ on $\omega$ of a Tarot 1755 propeller.

In Figure 12 the radial distributions of the angle of attack for both the minimum and maximum values of the angular velocity are presented.

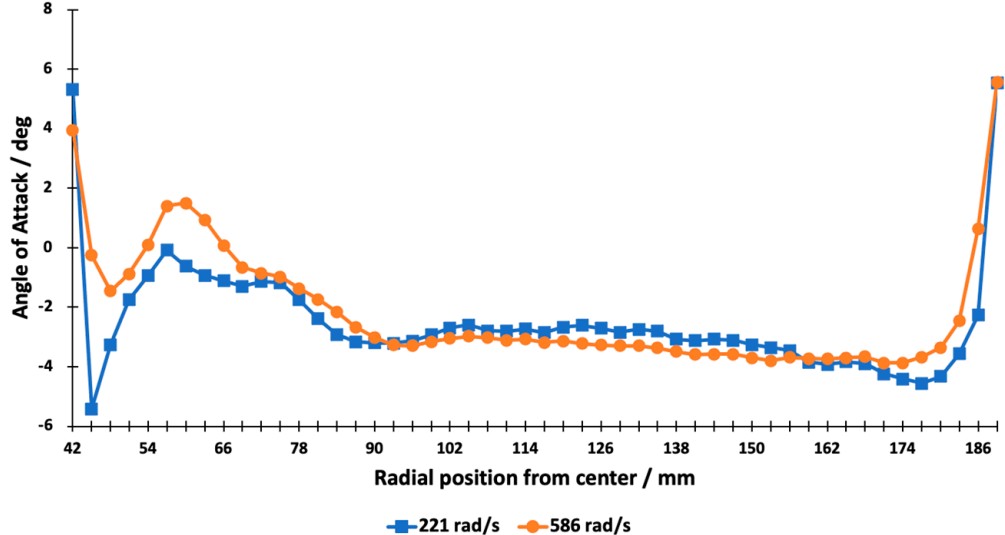

**Figure 12.** Radial distributions of angle of attack of 0.75 R blade section at minimum and maximum values of $\omega$.

Figure 13 presents the dependence of the effective lift coefficient and drag coefficient for the blade section at 75% of the propeller radius on the angular velocity and the Reynolds number of that particular blade section.

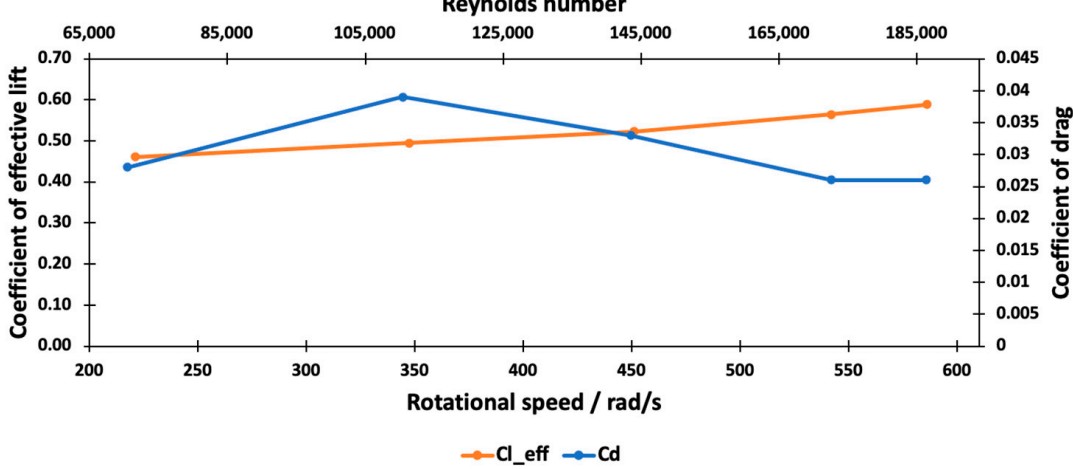

**Figure 13.** $C_{Leff}$ and $C_D$ for 0.75 R blade section against $\omega$ and *Re* for a Tarot 1755 propeller.

For comparison, another push-type static-use-oriented carbon-composite T-Motor FA20.2*6.6 propeller (20.2″ in diameter and 6.6″ in geometric pitch) was tested. This propeller uses a more advanced airfoil with a thin trailing edge. Its 0.75 R section has 8% of the relative thickness and 3.5% of the relative camber. The tip modifications used herein are blade tips bent toward the blade face forming approximately 20 mm long winglets with a positive sweep. The planform, side view, and 0.75 R airfoil are presented in Figure 14.

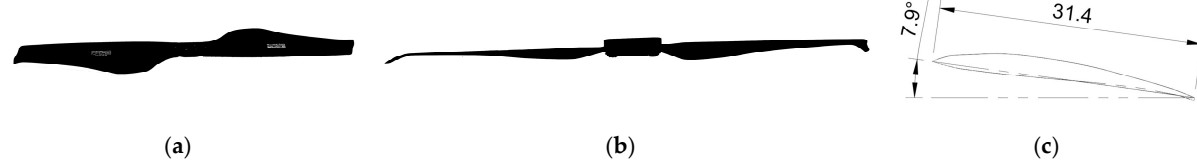

(**a**)                  (**b**)                  (**c**)

**Figure 14.** Planform (**a**), 83-degree side projection (**b**), and 0.75 R airfoil of a T-Motor FA20.2*6.6 propeller (**c**).

The radial distribution of the blade angle and cord length is presented in Figure 15.

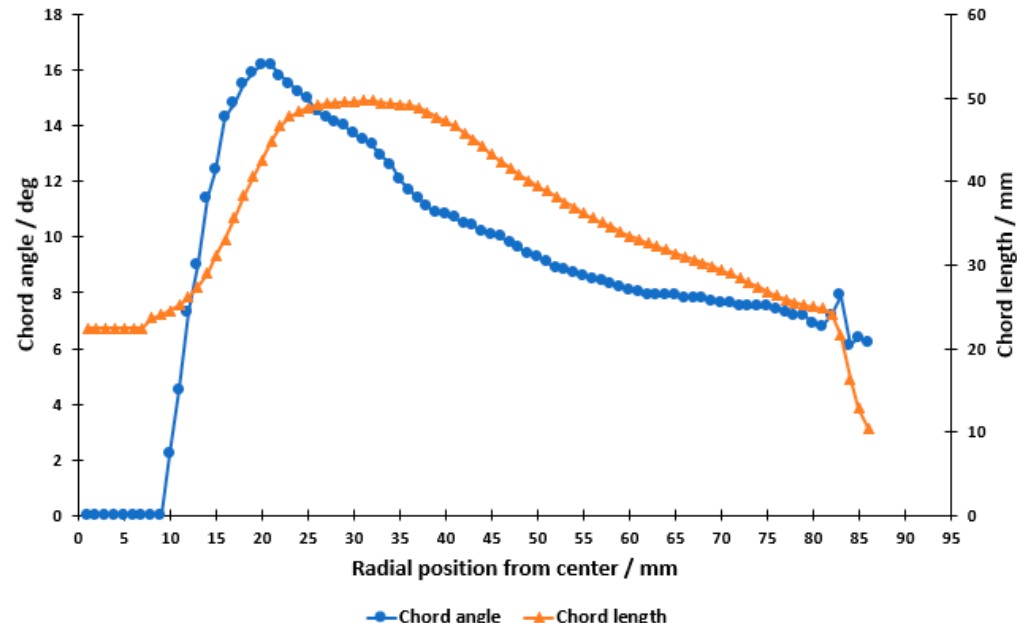

**Figure 15.** Radial distribution of cord length and blade angle of a T-Motor FA20.2*6.6 propeller.

The following graphs (Figure 16a,b) present the measured results for the T-Motor FA20.2*6.6 propeller in the same form as they were presented for the Tarot 1755 propeller.

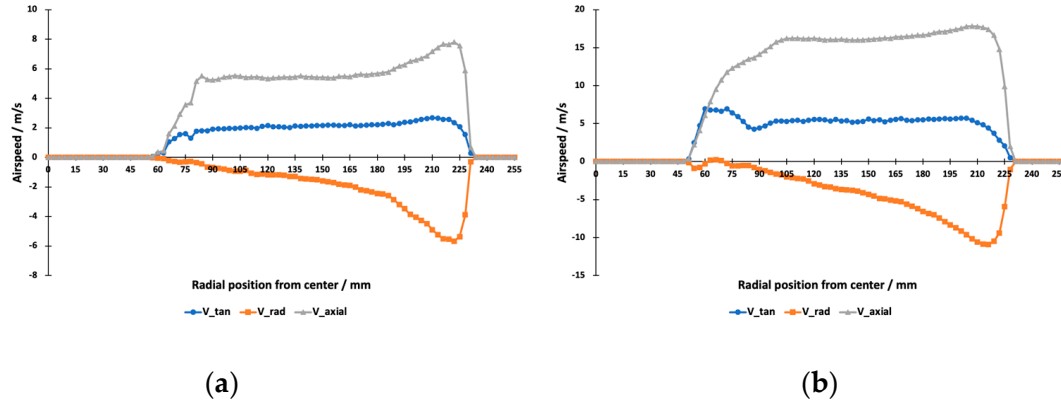

(**a**)                                        (**b**)

**Figure 16.** Radial distributions of $v_a$, $v_t$, and $v_r$ at approximately 2000 RPM (**a**) and approximately 6000 RPM (**b**) for a T-Motor FA20.2*6.6 propeller.

In Figure 17, the dependence of the thrust coefficient, power coefficient, and efficiency on the angular velocity is presented.

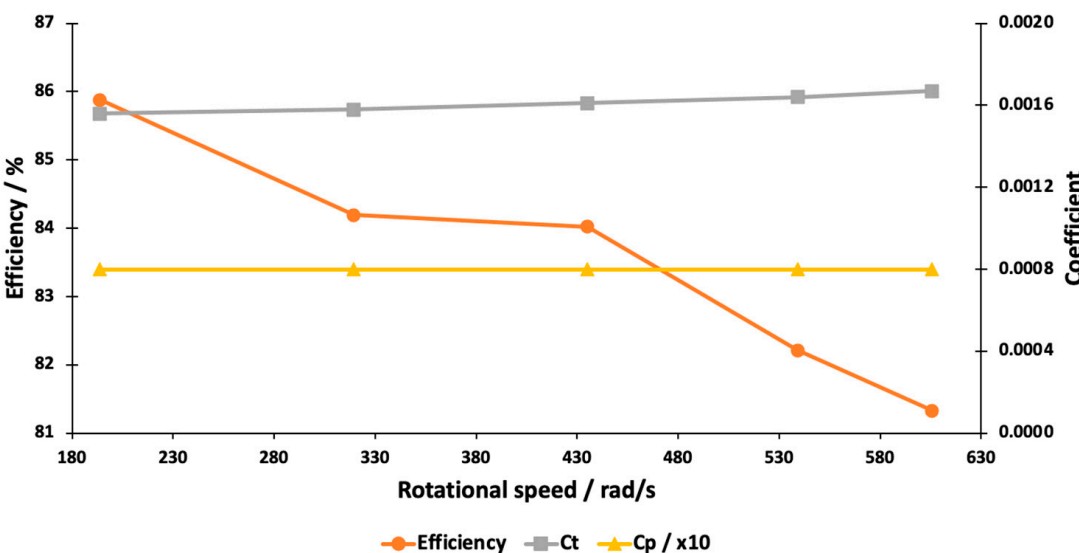

**Figure 17.** The dependence of $C_P$, $C_T$, and $\eta$ on $\omega$ for a T-Motor FA20.2*6.6 propeller.

In Figure 18 the radial distributions of the angle of attack for both the minimum and maximum values of the angular velocity are presented.

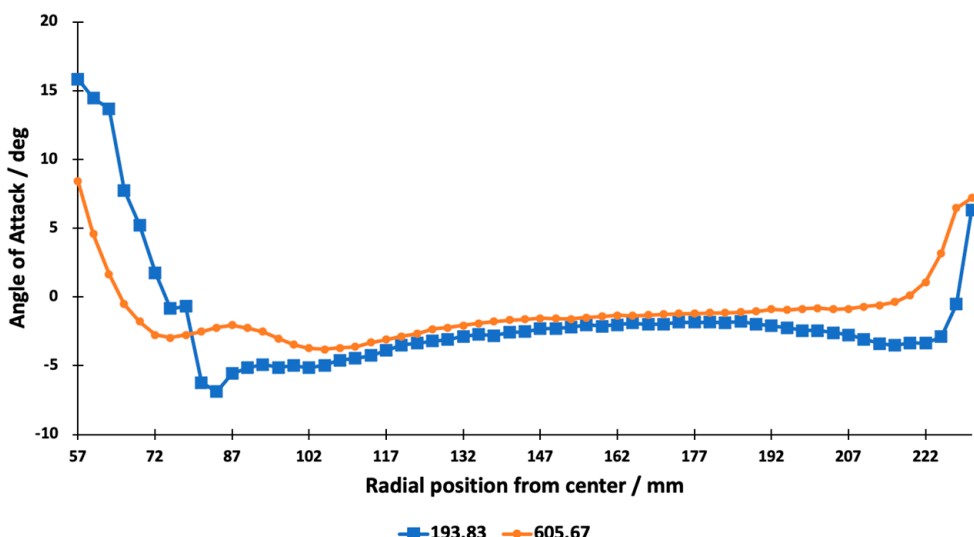

**Figure 18.** The dependence of angle of attack for 0.75 R blade section at minimum and maximum values of $\omega$.

In this paper, the measured and calculated parameters presented were chosen to demonstrate some possibilities of this method of investigation. Depending on the concrete aim of the investigation, many other parameters and characteristics can be measured using slipstream monitoring together with thrust and torque measurements.

## 5. Discussion

The two propellers that were tested are both oriented for hovering or nearly hovering flight mode. For this reason, these propellers are of quite a low geometric pitch (approximately 30% of the diameter) and highly cambered airfoil. Despite many similarities between the propellers, there are some significant differences including in efficiency, which could be called a basic characteristic of a propeller. The basic behaviors of the components of the induced velocity along the propeller radius are quite similar for both propellers. As

for differences, the Tarot 1755 propeller has a tendency for a higher $\frac{v_r}{v_a}$ ratio at the position of the axial speed maximum (66% for the Tarot 1755 propeller vs. 46% for the T-Motor FA20.2*6.6 propeller). The radial component of the Tarot 1755 propeller (Tarot) tends to rise toward the axis of rotation, while for the T-Motor FA20.2*6.6 propeller (T-Motor), it remains nearly constant. The blade characteristic points also have similar distances from the propeller axes. These points are the starting point, cut-off point, maximum speed point, center of pressure, and center of power.

The starting point is the point closest to the axis where the slipstream airflow becomes detectable (at an axial velocity of over 1 m/s). This point is located at 0.18 R for the Tarot propeller and 0.21 R for the T-Motor propeller.

The cut-off point is the point near the blade tip where the slipstream airflow drops below the detection level due to tip vortices. The position of this point is 0.88 R for Tarot and 0.89 R for T-Motor.

The maximum speed point is the blade section that generates the highest value of axial velocity. For both propellers, this section lies quite close to the break-off point, at 0.81 R for Tarot and 0.83 R for T-Motor.

The center of pressure is the thrust center of the blade. The location of this point is calculated based on the assumption that the thrust generated by a blade section is proportional to the blade width and square of the axial velocity. This point is located at 0.57 R for Tarot and 0.51 R for T-Motor.

The center of power is the blade section from which the blade half toward the axis generates the same power as the blade half toward the tip. This location is calculated assuming the generated power is proportional to the blade width and cube of the axial velocity. This point is at 0.61 R for Tarot and 0.58 R for T-Motor.

All these characteristic points were estimated at approximately 6000 RPM for both propellers. The well-known and widely recognized reference blade section to characterize the whole propeller is the section at 0.75 R. In this paper, this conventional blade reference section is also used, although none of the characteristic sections of the tested propellers are even close to that distance value.

In the literature, the thrust coefficient and power coefficient are presented as the basic parameters to characterize the efficiency of small-scale propellers. In most cases, with an increasing Re number, an increase in $C_T$ and/or a decrease in $C_P$ is registered. For instance, static measurements carried out for a 9" propeller showed a 20% increase in $C_T$, with $C_P$ remaining constant as the Re number increased from $2 \times 10^4$ to $7 \times 10^4$ [14]. These tendencies are generally interpreted as the impact of the Re number on the propeller's performance. The basic trend here seems to indicate that the lower the Re number of a propeller (for the 0.75 R section), the lower the propeller's performance. Measurements performed in the present work with Re numbers ranging from $7 \times 10^4$ to $2 \times 10^5$ showed an increase in $C_T$ for both propellers, at 26% for Tarot and 7% for T-Motor. As for the power coefficient, the results were different. For T-Motor, the value of $C_P$ remained strictly constant, but for Tarot, a 36% increase in $C_P$ was found. This result shows that the performance of a propeller is not predictable simply via the Re number and may depend additionally on some other parameters of the propeller. The measurements of propeller efficiency ended in results one could call unexpected. For both propellers, a decrease in efficiency was found with a rise in the Re number. For Tarot, the efficiency dropped from 74% at approximately 2000 RPM to 67% at approximately 6000 RPM. The same tendency, although lower, was registered for T-Motor, with $\eta$ dropping from 86% to 81%. It should be pointed out that there is a notable difference between the values of $\eta$ for these two propellers, which correlates with the differences in the values of $C_T$ and $C_P$. At 6000 RPM, the $C_T$ of Tarot was approximately 37% higher than that of T-Motor, but the $C_P$ of Tarot was nearly twice as high as that of T-Motor. The behavior of $C_P$ with the changing Re number indicates once more that the Re number is not the only key parameter by which a propeller's performance is determined. In the case of T-Motor, we could see a decrease in $\eta$ with the increasing Re number, although a significant 7% increase in $C_T$ and

constancy in $C_P$ were detected. These results lead to the hypothesis that the efficiency of a small-scale propeller is strongly dependent on the behavior of the slipstream. The behavior of the slipstream, leaving from the plane of rotation, depends on three velocity components induced by the propeller. It is known that if a slipstream contracts, the axial velocity increases and so does thrust. However, the radial component measured in the plane of rotation does not elucidate what that component will be at greater distances downstream. In this work, no systematic monitoring of the slipstream at greater distances was carried out, but some qualitative measurements at 7 cm from the plane of rotation were conducted. These measurements showed that there were no detectable mass rate losses at that distance, but significant changes in the radial distributions of the three velocity components were observed. It was noticed that at the minimum speed of rotation (2000 RPM) the radial flow component nearly reached zero at 7 cm from the rotating propeller. This indicates a very weak contraction of the slipstream at 2000 RPM and confirms the results obtained for the mean axial speed ratio of 1.05 for Tarot and 1.04 for T-Motor. The mean axial speed ratios were calculated from the measured thrust and mean induced velocity values. The mean axial speed ratio at 6000 RPM was found to be 1.48 for Tarot and 1.52 for T-Motor. This indicates a stronger slipstream contraction at higher values of RPM, and this greater contraction was also confirmed by the presence of a considerably higher value of the radial velocity at 7 cm from the propeller. It is also clear that the rise in $C_T$ for both propellers cannot be explained only by the slipstream contraction. The rise in the mean axial speed ratios was much greater (approximately 50%) than the rise in the thrust coefficients of the propellers. According to [15], the axial velocity reaches its maximum value at approximately 0.7 R–0.8 R downstream from the plane of rotation. The radial distribution also changes significantly; from a distance of approximately R, the slipstream starts to expand, and the region of the highest velocity moves toward the axis. It is possible to determine the mean maximum velocity from slipstream measurements when monitoring the slipstream at maximum contraction and integrating the axial component over the effective cross-section of the slipstream. This procedure is similar to that for determining the mean induced velocity in the present paper, wherein the effective cross-section was considered to be the propeller disc area. Slipstream monitoring further downstream together with induced velocity measurements provides a method to determine the energy losses in the slipstream and determine the thrust of the propeller. In [15], this method was called a slipstream method, and it showed a precision of 5–15% in comparison with direct thrust measurements, with no losses taken into consideration.

As for the angle of attack, both tested propellers worked in the region of negative angles of attack. In the main working region of the blade (0.4 R to 0.8 R), the angle of attack was constantly near $-3$ degrees for Tarot and rose from $-3$ degrees to nearly 0 degrees for T-Motor. For highly cambered airfoils such as those of our propellers, the zero-lift angle of attack is nearly $-5$ degrees. A slight dependence of the angle of attack distribution on the RPM value was noticed, but no interpretation was found for that effect. It must be noted that the fast rise in the angle of attack near the cut-off point was not adequate and was caused by the calculation procedure. The angle of attack is calculated as the difference between the blade angle and helix angle. As the slipstream velocity (but not $\omega \cdot r$) drops to zero, the calculated helix angle will also drop to zero and the angle of attack will show the value of the blade angle. We do not have methods to determine the angle of attack in the region where the slipstream velocity cannot be detected. In theory, the value of $\alpha$ near the blade tips must be somewhere near $-5$ degrees to produce no axial velocity. As the present method of testing enables to us to determine the effective lift coefficients and drag coefficients of all the working blade elements, this was also performed, and the results are presented only for the 0.75 R blade section for both propellers. The effective lift coefficient means that the found lift coefficient is multiplied by the speed ratio $\frac{\overline{v_{max}}}{\overline{v_i}}$. This is necessary because the lift coefficient is determined from the measured slipstream velocity components that do not take into account the impact of the pressure rise in the slipstream due to contraction and the axial velocity rise. The distributions of $C_{Leff}$ and $C_D$

along the blade are quite smooth (constant) in the main working region (0.4 R to 0.8 R). A very interesting result that could also be considered unexpected was the fact that the drag coefficients of the T-Motor blade sections were approximately 75% higher than those of Tarot despite the much better efficiency of T-Motor. At the same time, the effective lift coefficients were nearly comparable but still a little higher for Tarot. A 21 % rise in $C_{Leff}$ with the increasing Re number was detected for both propellers. The dependence of $C_D$ on the Re number was more complicated, as seen in the graphs in Figures 13 and 19, but it showed a general tendency to decrease with the increasing Re number. The airfoil of T-Motor at 0.75 R was compared with the NACA 6404 airfoil in order to compare the efficiencies of those airfoils by comparing data from the Airfoil Database with the present experiment. NACA 6409 was found to be the most similar airfoil to the 0.75 R airfoil of T-Motor. Unfortunately, it was not possible to find a similarly described airfoil for Tarot because this airfoil seems to be unfinished and hard to compare with the described airfoils. In Table 1, the data on the T-Motor 0.75 R airfoil and NACA 6409 are presented. The corresponding data for Tarot are also included. The data in Table 1 are not of high precision and are only able to illustrate stronger tendencies and differences.

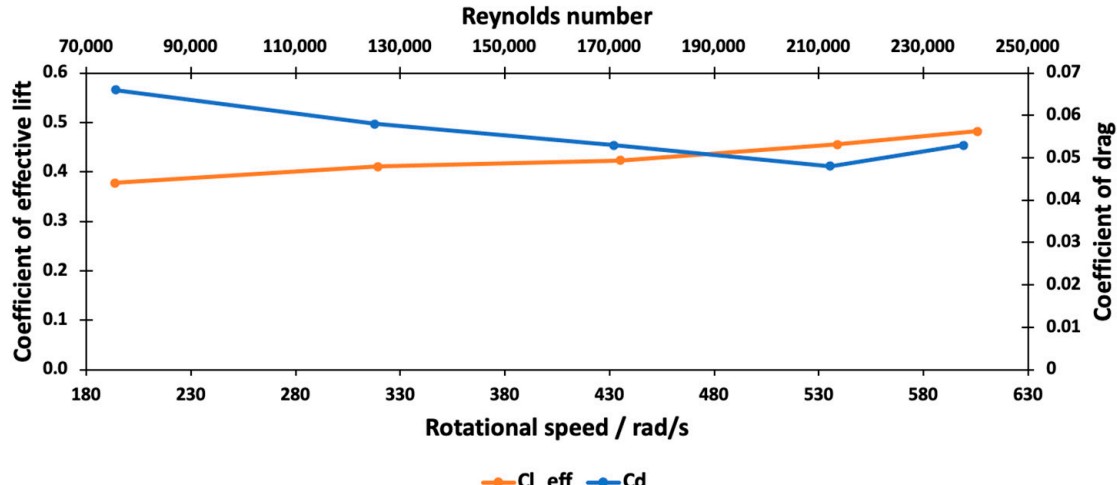

**Figure 19.** The dependence of effective lift coefficient and drag coefficient of 0.75 R blade section on $\omega$ and Re number.

**Table 1.** Comparison of airfoil efficiencies for T-Motor 0.75 R airfoil and NACA 6409 at similar angles of attack and Reynolds numbers, and data for Tarot 0.75 R airfoil.

| Airfoil | Re | $\alpha$ (deg) | $C_L$ | $C_D$ | $C_L/C_D$ |
|---------|------|------|------|------|------|
| NACA 6409 | 70,000 | −2 | 0.35 | 0.03 | 12 |
| T-Motor | 74,000 | −2 | 0.36 | 0.073 | 5 |
| NACA 6409 | 200,000 | −1.5 | 0.5 | 0.012 | 42 |
| T-Motor | 230,000 | −1.5 | 0.33 | 0.054 | 6 |
| Tarot | 70,000 | −3.5 | 0.41 | 0.041 | 10 |
| Tarot | 190,000 | −3.8 | 0.41 | 0.03 | 14 |

It is seen in Table 1 that if the lift coefficients of NACA 6409 and T-Motor airfoils are almost comparable at both values of Re numbers, the drag coefficients for the propellers are much (approximately 2.5 to 4 times) higher. This means that the airfoil working in a propeller blade does not show the same efficiency as predicted in the Airfoil Database. The data for Tarot show an even better airfoil efficiency even at lower angles of attack in spite of the fact that this particular airfoil is far from what one could call "aerodynamic", i.e., having

a constant thickness and blunt trailing edge (see Figure 8c). However, the better efficiency of T-Motor is correlated with the ratio of the thrust coefficient to the power coefficient, which is 15 for Tarot and 21 for T-Motor at Re numbers near 200 000. This shows that the better efficiency of T-Motor is not explainable by the efficiency of the airfoil used near the region of 0.75 R. It should also be mentioned that this kind of comparison does not take into account the fact that the effective airfoil is not the same as that directly measured for a propeller blade because of the radial airflow component in the plane of rotation. The radial component makes the effective airfoil thinner and decreases the camber. As the aim of the present paper was to introduce a method to quickly determine the efficiency of a propeller, not much attention was paid to all the possible applications of this method.

## 6. Conclusions

The presented method for small-scale propeller testing that combines thrust, torque, and angular velocity measurements with slipstream monitoring is a quick and effective way to obtain necessary data on propellers and to determine static efficiency. It enables us to study the dependence of propellers' performance on geometry, pitch, tip modifications, Re number, etc. It also enables us to carry out further investigations to find answers to the questions that arose during the present investigations. The most important question is the impact of the Re number on the performance of a small-scale propeller. The tests showed a decrease in efficiency with the rising Re number in spite of the rising values of the thrust coefficient. This leads to two main conclusions.

First, the thrust and power coefficients are not always the key parameters to determine the efficiency of a propeller, and second, the role of the Re number in a propeller´s efficiency is not yet clear. The presence of Re number effects has been proven in numerous works, but the impact of those effects seems to not be as trivial as the claim that the lower the Re number, the weaker the performance. To determine the real impact of the Re number, small-scale propellers of significantly different diameters at different angular velocities must be tested with the Re number remaining constant. This would enable one to exclude the Re number effects and find out the possible role of other parameters. In this paper, the slipstream was monitored closest to the trailing edge of the propeller to detect the induced airspeed most precisely. This method would also be successful for slipstream monitoring further downstream to investigate the behavior of a slipstream. This may be necessary to find out what factors determine the rate of slipstream contraction between the plane of rotation and the point of maximum axial velocity. It is also possible to investigate not only propellers but also devices such as ducted fans and jets of any origin because this would enable us to determine the axial mass rate at any point on a slipstream or jet and the mass rate losses along the stream. As all three components of the slipstream velocity are measured, it would be possible to detect the effect of different duct constructions to decrease slipstream whirling because the tangential velocity in a slipstream is quite high, approximately one-third of the axial velocity. Decreasing the tangential velocity may produce a significant effect on the performance of ducted fans. Currently, the only serious limitation of this method is its usability only in static conditions, i.e., not in advancing flow. The test stand is not intended for use in wind tunnels because, for that purpose, it must be have a fundamentally different construction. However, many investigations will be carried out on thrust-producing small-scale devices in static conditions.

**Author Contributions:** Conceptualization and methodology, J.S. and K.-E.U.; software, validation, and experiments, S.H.; writing—original draft preparation, J.S.; writing—review and editing, K.-E.U. All authors have read and agreed to the published version of the manuscript.

**Funding:** This research received no external funding.

**Data Availability Statement:** The results and experimental data can be provided upon request.

**Conflicts of Interest:** The authors declare no conflict of interest. Other parties had no role in the design of this study; in the collection, analysis, or interpretation of the data; in the writing of the manuscript; or in the decision to publish the results.

## Abbreviations

This paper uses SI units and symbols as follows:

| | |
|---|---|
| $M$ | Torque (Nm) |
| $T$ | Thrust (N) |
| $P$ | Input power (W) |
| $P_i$ | Induced power (W) |
| $C_T$ | Thrust coefficient |
| $C_p$ | Power coefficient |
| $L$ | Lift force (N) |
| $D$ | Drag force (N) |
| $C_L$ | Lift coefficient |
| $C_D$ | Drag coefficient |
| $v_v$ | Volumetric rate (m/s) |
| $v_i$ | Induced velocity (m/s) |
| $\overline{v_i}$ | Mean induced velocity (m/s) |
| $v_a$ | Axial component of the induced velocity (m/s) |
| $v_r$ | Radial component of the induced velocity (m/s) |
| $v_t$ | Tangential component of the induced velocity (m/s) |
| $v_0$ | Freestream flow velocity (m/s) |
| $v_T$ | Total airspeed (m/s) |
| $v_L$ | Downwash velocity (m/s) |
| $v_D$ | Drag velocity (m/s) |
| $J$ | Advance ratio |
| $\eta$ | Propeller efficiency |
| $D$ | Propeller diameter (m) |
| $R$ | Propeller radius (m) |
| $r$ | Radial parameter (m) |
| $\alpha$ | Angle of attack (deg) |
| $\beta$ | Helix angle (deg) |
| $\gamma$ | Blade angle (deg) |
| $\omega$ | Angular velocity (rad/s) |

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
