# Peer review of "Determining the Efficiency of Small-Scale Propellers via Slipstream Monitoring"

_drones, doi:10.3390/drones7060381_

Round 1

Reviewer 1 Report

The purpose of the article was to introduce a method for empirically determining the static efficiency of small propellers by comparing output and input power, where output power is determined by measured thrust and average induced speed. The paper presents the author's method based on a built stand for measuring the static efficiency of small propellers. However, the authors did not sufficiently define what is meant by the term small propellers. This needs to be supplemented with the parameters of the propellers for which the method is dedicated. The authors also did not sufficiently explain why other methods are not suitable for analyzing these propellers. So the scientific contribution here is not clearly defined. This needs to be supplemented. In terms of editorial comments, the abstract is a duplication of content from the article's introduction. The content in the article must not be repeated. The abstract therefore needs to be re-edited. All formulas given in the text should be supplemented with units.

Author Response

We modified the article accordingly:

Question 1. what is meant by the term small propellers:

  1. Introduction on page 2 was modified: "The definition of the term “Small Scale Propeller” also is determined by the fact whether the main parts of a propeller (as a rule blade sections near 75% of the propeller radius) are working either at lower or higher Reynolds numbers than the critical. Propellers working at Reynolds numbers lower critical are often called Small Scale Propellers and so are the propellers investigated in the present paper (Reynolds numbers between 70, 000 and 200, 000)."

Question 2: explain why other methods are not suitable for analyzing these propellers

In the introduction, the current wording states: The problem is that useful power or output power is not measured empirically and only thrust and power coefficients are compared to evaluate a propeller´s performance. Sometimes the ratio of those coefficients is used to evaluate the performance of a propeller. Thrust is the only output characteristic in these cases. The induced velocities for a propeller may be calculated from the momentum theory (13), but for the reasons mentioned above those calculations may not have enough precision at low Reynolds numbers. This leads to a need for direct measuring of the induced velocity to define the static efficiency of a propeller via that parameter.

The abstract was modified with the wording:

There have been several previous studies on static and advancing propeller performance. In these studies, when determining the efficiency of a static propeller thrust and power coefficients are still most commonly compared to evaluate a propeller´s performance. Sometimes the inducted velocities are calculated from the momentum theory. As the small scale propellers work on very low Reynolds (Re) numbers below 500 000, the flow type transition and boundary layer separation make it very hard to predict the actual efficiency of the propeller in static mode. 

3. Question: The abstract, therefore, needs to be re-edited.

The abstract was re-edited. Please see the attached version of the article

Reviewer 2 Report

General comments:

The selected topic is innovative and the working procedure is well laid out with a good presentation of the difficulties of propeller aerodynamics at low Reynolds number. The experimental test is well founded and is able to obtain complete information on the behaviour of the velocities induced in the propeller wake. The results are consistent with and complement traditional measurements of thrust coefficients, torque and revolutions to obtain new and detailed conclusions. The conclusions are detailed and consistently connected with the results. However, several of the conclusions are not definitively closed.

The reviewer proposes that, in the Discussion section, the influence of the aerodynamic efficiency of the airfoils used on both propellers as a function of the angle of attack should be assessed. For this purpose, it is necessary to have the curves of the experimental aerodynamic coefficients of the two airfoils for the angles of attack and the working Reynolds numbers considered.

Detailed comments:

- For the mean induced velocity it is proposed to use the symbol vi always with the stripe at the top, instead of the one used (witch sometimes have it at the top and sometimes at the bottom). This decision should be reviewed and standardised throughout the article in lines 88, 121, 217, 278 and 544.

- In order not to confuse the symbol d used for the chord with the differential sign d, it is proposed that the chord be written as d (italic) and the differential sign as d (roman). Revise lines 189, 211, 215, 251 and 256,

Author Response

Question 1: The influence of the aerodynamic efficiency of the airfoils used on both propellers as a function of the angle of attack should be assessed:

Discussion part was modified with a comparison table (Table 1) and with a added paragraph: "

It is seen from Tab.1 that if the lift coefficients of NACA 6409 and T-Motor airfoils are almost comparable at both values of Re numbers, the drag coefficients for the propeller are much (about 2.5 to 4 times) higher. This means, that the airfoil working in a propeller blade does not show the same efficiency as predicted by Airfoil Database. The data for Tarot show even better airfoil efficiency even at lower angles of attack in spite of the fact that this particular airfoil is far from what one could call “aerodynamic”, i.e. having constant thickness and blunt trailing edge (see Fig. 8-c). However, the better efficiency of T-Motor is correlated with the ratio of thrust coefficient and power coefficient which is 15 for Tarot and 21 for T-Motor at Re numbers near 200 000. This shows that the better efficiency of T-Motor is not explainable by the efficiency of airfoil used near the region of 0.75R. It should also be mentioned, that this kind of comparison does not take into account the fact, that the effective airfoil is not the same as directly measured for a propeller blade because of the radial airflow component in the plane of rotation. The radial component makes the effective airfoil thinner and decreases the camber. As the aim of the present paper is to introduce a method to quickly determine the efficiency of a propeller, not much attention was paid to all the possible applications of this method."

Queston 2: For the mean induced velocity it is proposed to use the symbol vi always with the stripe at the top, instead of the one used

Noted, corrected (please see the attached article)

Question 3  In order not to confuse the symbol d used for the chord with the differential sign d, it is proposed that the chord be written as d (italic) and the differential sign as d (roman)

We corrected the formating and used roman for chord and differential in italic (please see the attached article).

Round 2

Reviewer 1 Report

All my comments have been taken into account in the second version of the manuscript